# OpenReview forum: "DAG-Jailbreak: Enhancing Black-box Jailbreak Attacks and Defenses through DAG Dependency Analysis"
_ICLR.cc/2025/Conference — Submitted to ICLR 2025_

### Official Review · Reviewer_8r3u · 2024-10-24

**Soundness:** 3
**Presentation:** 3
**Contribution:** 3
**Rating:** 6
**Confidence:** 3

**Summary:**

This paper presents the DAG-Jailbreak framework, which employs Directed Acyclic Graph (DAG) analysis to enhance both jailbreak attacks and defenses for LLMs. The framework consists of three key components: DAG-Attack, which optimizes attack strategies using mutation and adversarial generation methods; DAG-Defense, which introduces a Mixture-of-Defenders approach to improve the generalizability and effectiveness of defenses; and DAG-Evaluation, which assesses the success of attacks and defenses, incorporating the concept of Jailbreak Hallucination to identify irrelevant responses. Experimental results demonstrate the framework's ability to significantly improve both attack success rates and defense robustness across multiple LLMs.

**Strengths:**

1. The paper presents a comprehensive DAG-Jailbreak framework that includes attack strategies (DAG-Attack), defense mechanisms (DAG-Defense), and evaluation methods (DAG-Evaluation).
2. It provides thorough experimental validation across various LLMs, such as GPT-3.5, GPT-4, and LLaMa-2, showing significant improvements in attack success and defense effectiveness over baseline methods.
3. The framework's adaptability and ability to integrate new methods make it highly scalable.

**Weaknesses:**

1. The DAG formulas lack detailed explanations, particularly regarding how they ensure global optimization and eliminate redundant paths.

2. Experimental parameters like temperature, top-k, and top-p are minimally described. More detail would improve the transparency and reproducibility of the experiments.

3. The concept of Jailbreak Hallucination needs a clearer distinction from typical LLM hallucinations. Further clarification would enhance understanding.

4. Minor grammatical issues, such as "semantical similarity" instead of "semantic similarity," slightly affect the paper's polish.

**Questions:**

1. In DAG-Evaluation, how are Keywords Matching, Binary Classifier, and LLM-as-a-Judge fairly compared, especially in handling Jailbreak Hallucination, and what are their key differences?

2. Given the complexity of manual DAG dependency analysis, how can global optimality of attack combinations be ensured, and is there proof of this?

3. Could you provide details on the computational overhead?

4. Could you explain more about the practical application of the DAG-Jailbreak framework in real-world scenarios?

---

> ### Comment · Reviewer_8r3u · 2024-11-26
>
> I would like to thank the authors for their response. However, upon further reflection, I have concerns about the validity and persuasiveness of the problem that the paper addresses.  I will keep my current rating.

---

### Official Review · Reviewer_LSXh · 2024-10-28

**Soundness:** 3
**Presentation:** 1
**Contribution:** 3
**Rating:** 5
**Confidence:** 2

**Summary:**

The paper proposes DAG-Jailbreak, a framework for combining existing LLM jailbreaking attack and defense methods to achieve stronger overall performance. Both the attack and defense methods are empirically evaluated under LLM-as-Judge with LLaMa-3 as the judge. The attack method is shown to be significantly more effective than several recent baselines from the literature across a wide range of LLMs. The defense method is shown to reduce the jailbreak hallucination rate (responding to the jailbreak prompt with non-harmful content), improving the correct refusal rate, although sometimes at the cost of increasing the jailbreak success rate.

**Strengths:**

- Important problem domain.
- Strong and comprehensive empirical evaluation results.
- Highlights commonalities among LLM jailbreaking works.

**Weaknesses:**

Despite the impressive evaluation results, this paper fails to sufficiently communicate its core contribution: a general method for integrating multiple LLM jailbreaking attack and defense techniques. There are two critical gaps in understanding that would prevent me from reproducing this paper's work and consequently prevent me from accepting it:

1. The dependency analysis is not formalized, despite claims that it can be automated. The process of creating a dependency graph from a set of independently developed jailbreaking techniques is not trivial and is not effectively described by the paper. Without further explanation of the dependency analysis process, I would be unable to reproduce the dependency graphs presented in Figure 1 and Figure 2.

2. The process of creating a combined attack or defense method from a dependency graph is unclear and represents a nontrivial engineering challenge when independently developed techniques do not adhere to any set interface. Without a detailed discussion of how the dependency graph is applied to generically compose existing techniques, I would be unable to reproduce the attacks and defenses presented, even with the dependency graphs from Figure 1 and Figure 2.

**Questions:**

- Can you algorithmically describe the process used to determine the dependencies shown in Figure 1 and Figure 2?
- Can you algorithmically describe the technique composition process used to produce the attacks and defenses presented in the evaluation from the dependency graphs in Figure 1 and Figure 2?
- Are DAGs used in DAG-Evaluation? If not, the name may be misleading.

Minor Writing Recommendations:

As may be clear from the rest of the review, my complaints are largely centered on the organization and presentation of ideas. Towards improving the organization of the paper, it may be best to present DAG-Attack and DAG-Defense separately, each with their own background and evaluation. This organization choice would reduce the mental load on the reader.

Further, there are several references to global/local optimization without formalizing the optimization target or clarifying what is meant by global and local. The abstract and introduction both refer to "global algorithms", which are not concretely introduced until section 3.2 and 3.3, which are halfway through the paper. By introducing the specific global algorithms used to systematize the current literature earlier, design insights of DAG-Jailbreak could be made clearer while also allowing the paper to be better understood in a single pass. Formalizing the inputs and outputs of each stage of the attack and defense algorithm decomposition may help future LLM jailbreaking works consistently apply your framework.

The paper demonstrates state-of-the-art results in an important problem domain and is well-situated in the literature, but does not provide enough information to reproduce the work.

---

### Official Review · Reviewer_S31b · 2024-11-02

**Soundness:** 3
**Presentation:** 2
**Contribution:** 2
**Rating:** 6
**Confidence:** 4

**Summary:**

This paper presents Directed acyclic graphs (DAG) approach to interact with large language models (LLMs) and elicit a jailbreak condition This paper introduces DAG frameworks. This paper introduces DAG-Jailbreak, a novel framework leveraging Directed Acyclic Graph (DAG) dependency analysis to construct three comprehensive, automated, and logical frameworks for robust jailbreak attack, defense, and evaluation methodologies.

**Strengths:**

1. The paper is very detailed in experimentation. For showing the effectiveness of the attacks and defense.
2. The approach is tested on both open-source and Closed-source LLM.

**Weaknesses:**

Algorithmic Detail: Key steps, like mutation, selection, and adversarial generation, are described conceptually but lack specific algorithms or pseudocode, making it hard to replicate exactly.

Dependency Management: The DAG structure’s dependencies between attack components are not clearly defined, leaving ambiguity in handling conflicts or prioritizing nodes.

Hyperparameters and Configuration: Parameters for mutation rates, prompt scoring, and component-specific configurations are not provided, which would require experimental tuning.

Evaluation and Termination Criteria: The paper lacks clear metrics for attack success and stopping conditions, which are crucial for implementing the iterative process efficiently.

Lack of Algorithmic Details
Mutation and Selection Mechanisms: While the paper outlines stages like seed initialization, selection, and mutation for DAG-Attack-Mut, it does not provide the actual algorithms or pseudocode. For instance, it mentions using techniques like AutoDAN-GA and GPTFuzzer but lacks specifics on parameter settings, mutation strategies, or how to handle dependencies between mutations.
Adversarial Generation Process: The DAG-Attack-Gen process is also described in broad strokes, such as using an adversarial LLM for red-teaming, but it doesn’t specify how the adversarial model should be trained or configured to generate optimized jailbreak prompts. Key details on prompt design, feedback loops, or specific evaluation metrics are missing.

**Questions:**

1. The Prompt generation process is not clearly Explained.

**Details Of Ethics Concerns:**

Not needed

---

### Official Review · Reviewer_5gP8 · 2024-11-02

**Soundness:** 2
**Presentation:** 3
**Contribution:** 1
**Rating:** 5
**Confidence:** 4

**Summary:**

The paper introduces DAG-Jailbreak, a framework improving black-box jailbreak attacks and defenses for LLMs. DAG-Jailbreak leverages Directed Acyclic Graph (DAG) dependency analysis to create three key components: DAG-Attack, which builds effective attack strategies, DAG-Defense, a novel defense mechanism using a Mixture-of-Defenders; and DAG-Evaluation, which incorporates Jailbreak Hallucination and a two-stage assessment framework for evaluating LLM outputs. Experiments demonstrate that this method enhances attack effectiveness, improves defense generalizability, and refines evaluation practices.

**Strengths:**

- This work proposed the use DAG dependency analysis to facilitate the enhanced design of jailbreak attack/defense/evaluation, which gives a broader view of the field to study
- A comprehensive study of black-box attacks including both mutation and adversarial generation-based algorithms is provided, and the DAG-attack designs show promising improvement toward individual attack methods on most models.
- A comprehensive study of black-box defense is conducted based on the mixture-of-defense mechanism, that assigns different defense methods to specialized defense. The results show good defense performances and generalizability of the design.
-  A often overlooked metric, namely Jailbreak Hallucination is proposed in the evaluation, which is further refine the evaluation study of jailbreak.

**Weaknesses:**

- Though the DAG analysis covers a diverse set of representative methods, it highly depends on the human efforts to conduct the analysis. The ensemble of different methods for each group (e.g., DAG-Attack-Gen) did not employ any automatic pipeline to improve each decomposed component/verify the improvement of each replacement. For example, DAG-Attack-Mut got even worse JR on GPT-4 and Claude models, which indicates that the design did not get the expected improvement from the analysis.
- As randomness is introduced by setup 'temperature to 0.7, top-p to 0.7, and top-k to 50', the reproductivity is not under strong control. And given the limited number of evaluated data samples, it lacks statistical analysis such as a confidence interval to show the improvement is not marginal and random.
- As a defense design, utility preservation is not even considered for the evaluation of DAG-Defense. High defense effectiveness may sacrifice the general utility, especially when this is an ensemble design. Also, the cost of defense is not discussed and evaluated. When multiple defense are stacked together, the cost the deploying the design will also increase significantly.

**Questions:**

- In the design of defense, it assumes the access to a justification whether the prompt is malicious or benign. When the prompt is judged as malicious, why does it still need to be served through a set of defense components instead of simply a refusal answer?
- The evaluation metrics are confusing. In particular, three metrics, JR, HR and AR are evaluated, but 'JR and AR as the main
criterion' for attack and defense evaluation, then what is the point of proposing and evaluating HR metric? By intuition, I think HR cannot be regarded as successful jailbroken, but close to an aligned state that did not provide harmful information. Therefore at least on the evaluation on jailbreak defense, HR+AR is a more reasonable metric than AR.

---

### Meta-Review · Area_Chair_2Y5w · 2024-12-16

**Metareview:**

The paper proposes a DAG-based framework to construct jailbreaks by combining different sub-components.
The reviewers generally agreed that the results of the jailbreaks were good, but raised concerns about the automatability of the approach, as well as the potential impact on utility of the proposed defense.
Overall, I believe this paper requires more work to be accepted.

**Additional Comments On Reviewer Discussion:**

A core concern raised in some reviews was the utility impact of the proposed defense, which was not evaluated.
The authors' rebuttal included one experiment on MMLU, from which it was not clear whether the drop in utility was due solely to detecting some questions as unsafe, or whether the remaining safe questions also saw a drop in utility.
Overall, further utility tests would be required to conclude that the defense has a minimal impact.

---

### Decision · Program_Chairs · 2025-01-22

Reject